# Characterization of Mode I and Mode II Interlaminar Fracture Toughness in CNT-Enhanced CFRP under Various Temperature and Loading Rates

**DOI:** 10.3390/nano13111729

**Published:** 2023-05-25

**Authors:** Burak Yenigun, Muhammad Salman Chaudhry, Elli Gkouti, Aleksander Czekanski

**Affiliations:** Department of Mechanical Engineering, York University, Toronto, ON M3J 1P3, Canada; byenigun@yorku.ca (B.Y.); salman01@yorku.ca (M.S.C.); gkoutiel@yorku.ca (E.G.)

**Keywords:** carbon nanotubes (CNT), Mode I interlaminar fracture toughness, Mode II interlaminar fracture toughness, double cantabile beam (DCB), end-notched flexure (ENF)

## Abstract

This study investigates the influence of temperature and loading rate on the Mode I and Mode II interlaminar fracture behavior of carbon-nanotubes-enhanced carbon-fiber-reinforced polymer (CNT-CFRP). CNT-induced toughening of the epoxy matrix is characterized by producing CFRP with varying loading of CNT areal density. CNT-CFRP samples were subjected to varying loading rates and testing temperatures. Fracture surfaces of CNT-CFRP were analyzed using scanning electron microscopy (SEM) imaging. Mode I and Mode II interlaminar fracture toughness increased with increasing amount of CNT to an optimum value of 1 g/m^2^, then decreased at higher CNT amounts. Moreover, it was found that CNT-CFRP fracture toughness increased linearly with the loading rate in Mode I and Mode II. On the other hand, different responses to changing temperature were observed; Mode I fracture toughness increased when elevating the temperature, while Mode II fracture toughness increased with increasing up to room temperature and decreased at higher temperatures.

## 1. Introduction

Carbon-fiber-reinforced polymer (CFRP) composites are widely utilized as structural materials in engineering applications due to their exceptional properties, such as high specific strength, corrosion resistance, high stiffness, and fatigue resistance [1,2,3,4]. Carbon fiber reinforced in an epoxy polymer matrix is known to exhibit a high in-plane tensile strength (up to 1400 MPa); however, applications where the peel stresses exceed the matrix’s tensile strength can result in delamination failure within composite layers [5,6,7]. In real-world applications, harsh environmental conditions and loading rates can also contribute to premature delamination. In order to improve interlaminar strength, it is necessary to understand the propagation of cracks and characterize its critical energy release rate under various environmental and loading conditions [8,9].

The delamination of carbon-fiber-reinforced polymer (CFRP) components can result in significant material damage and shorten their service life. Several approaches have been proposed to improve the interlaminar fracture toughness of these components, including stitching, braiding [10], matrices toughening [11,12], and z-pinning in the through-thickness direction [13]. However, these techniques may be expensive, difficult to implement, or negatively affect the performance of individual fibers. As an alternative solution, using nanofillers with superior mechanical properties to strengthen the carbon fiber polymer matrix and enhance fiber-to-matrix interaction has been proposed [14,15]. Various nanofillers, such as ceramic nanotubes, graphene nanotubes, carbon nanotubes, and natural fiber, have been used in engineering applications [16]. Several types of carbon fiber, such as quartz and glass fibers, have been utilized due to their respective advantages [17,18]. Studies have shown that composite laminates containing CNT in the delamination plane exhibit enhanced fracture toughness [19,20,21]. Novel methodologies have been devised to enhance the alignment of CNTs by grafting them onto carbon fibers. However, achieving and sustaining a uniform distribution of CNTs within a matrix poses significant challenges [22,23]. CNTs offer versatile structural properties such as high strength, stiffness, and low weight, and can be found in single-walled and multi-walled forms [24,25,26]. To obtain a uniform dispersion of CNTs, methods such as sonication, stirring, spraying, and calendaring are utilized. To address any potential issues related to imperfect CNT dispersion that may lead to a decrease in fracture toughness, several approaches have been developed to ensure the attainment of enhanced mechanical properties [27,28,29]. 

Investigations have shown that CFRPs exhibit interlaminar strength sensitive to loading and operating temperature. Another factor with a significant impact on those composite’s mechanical responses is the amount of CNTs. However, this relationship is non-linear, and it is necessary to characterize the optimal amount of CNTs [23]. An assessment was conducted to examine the individual impact of various factors, including CNT loading, temperature, and loading ratio, on the interlaminar fracture toughness in both Mode I and Mode II. This investigation encompassed different cases of CNT fabrication. Notably, in the case of woven CFRP, the interlaminar fracture toughness exhibits variation with different CNT loadings, reaching its maximum value at 1 g/m^2^ [23]. Furthermore, the influence of loading rate was explored by subjecting varying CNT film layers to static and dynamic loading (high strain rate) conditions [30]. It was observed that the fracture toughness displayed a monotonic increase with an increasing number of CNT film layers under dynamic loading. However, a slight decrease in fracture toughness was observed under static loading conditions. In determining the optimum areal density of CNTs, individual impact factors have been examined, such as loading rates. It is important to note that the investigation focused solely on Mode I fracture toughness. The effect of those factors (CNT loading, temperature, and loading ratio) at the same time on the fracture toughness has previously been investigated only for composites containing electrospun CNTs with different areal densities [31]. Outcomes showed that Mode I and Mode II fracture toughness increased with increasing areal density of CNTs.

Furthermore, at various operating temperatures (−40 °C, room temperature, and +80 °C) carbon fiber composites reinforced with multi-walled carbon nanotubes have demonstrated enhancement in flexural strength up to 10% [32]. It was also observed that at −40 °C, higher localization of the impact event resulted in the lowest peak force and a maximum displacement higher than room temperature. In another study, unidirectional glass fibers were toughened with the addition of CNT-infused epoxy films to optimize the interlaminar toughness in between delaminating layers. Specimens were placed under a low-temperature circulation treatment in advance, and another group was placed under RT as a control without treatment [3]. As a result of the performed temperature cycling, Mode I interlaminar fracture toughness values were lower compared to those without temperature cycling. The alternating stresses caused by fiber-reinforced plastics after low-temperature cycle treatment are mostly present at the fiber matrix level. In the presence of different coefficients of thermal expansion (carbon fiber, epoxy resin, and CNTs), external and mutual restrictions prevent expansion and shrinkage, which results in residual stress. However, the impact on the fracture toughness when CNT-CFRP is exposed to different-than-room temperature was only examined in a limited loading rate range.

The primary goal of the current study was to analyze the CNT-CFRP fracture behavior when subjected to various impact factors, such as CNT areal density, temperature, and loading rate. Initially, an optimization process was performed to determine the optimum CNT areal density so that the interlaminar strengthening effect is maximum. Using the maximum improvement in interlaminar fracture toughness at room temperature, Mode I and Mode II fracture toughness was further tested under different temperatures and loading rates. Scanning electron microscopy (SEM) was used to conduct a detailed analysis of the fracture surface of CNT-DFRP, revealing that the interplay between these factors has a significant and noticeable effect.

## 2. Materials and Methods

### 2.1. Interlaminar Fracture Toughness in Mode I and Mode II

The fracture toughness of composite materials is quantified through the measurement of the cracking strain energy release rate, which is determined through interlaminar fracture tests [33,34,35]. Interlaminar fracture tests are named based on Mode I and Mode II loading modes. In the Mode I test, also known as the double cantabile beam test (DCB), loading is applied in a direction normal to a crack plane. In the Mode II test, also known as the end-notched flexure (ENF) test, loading is applied as a three-point bending, parallel to the crack plane and perpendicular to the crack front. Different loading modes are shown in Figure 1.

### 2.2. Materials

Unidirectional (UD) carbon fiber–epoxy prepreg laminates, VTC401-C150-24, were supplied by SHD Composites. These UD laminates have an areal weight of 150 g/m^2^ and a 38% epoxy resin content by weight. Multi-walled carbon nanotubes (MWCNTs) with an average diameter of 9.5 nm, NC7000, were used as nanofillers. Ethanol was used to make a spraying solution of 0.2 wt% CNT. A non-adhesive polytetrafluoroethylene (PTFE), with a required thickness of 13 μm according to ASTM D5528-01, was used as an insert in the mid-plane of the laminate during layup to form an initiation crack for the delamination.

### 2.3. Samples Preparation

First, the CFRP prepregs were cut with dimensions of 150.81 mm × 152.4 mm and 150.81 mm × 203.2 mm for Mode I and Mode II interlaminar fracture tests, respectively. Then, PTFE was inserted on the surface of the mid-layer. At the same time, CNT and Ethanol solutions were prepared at 0.2 wt%. A mechanical stirrer and ultrasonic cleaner, shown in Figure 2a, were operated for 75 min to disperse CNT effectively in ethanol. Once the solution was prepared, it was sprayed on the surface of the mid-layer using an airbrush. After drying, the other half of the lamina, without CNT or insert, was attached to CNT-sprayed and PTFE-inserted lamina (Figure 2b). Then, multiple samples and glass molds were placed on top of each other to cure multiple samples efficiently. Once multiple glass molds were stacked, the glass mold and uncured laminates were wrapped in a breather fabric, as shown in Figure 2c. Since the wrapping procedure was completed, the glass mold with the uncured samples was placed inside the vacuum bag and sealed entirely on all sides. Slight vacuum pressure was applied to the vacuum bag to prevent air leakage (Figure 2d). Finally, the vacuum bag was placed into the oven to cure samples for 1 h at 100 °C. After 1 h of curing, the temperature was increased to 135 °C, and samples were cured for two more hours.

Afterward, the samples were cooled to room temperature and cut with a water jet according to the ASTM standards. Mode I and Mode II interlaminar fracture tests sample dimensions are shown in Figure 3.

### 2.4. Experimental Procedure

An MTS Universal test machine with 500 N and 10 kN load cells was used to perform DCB and Mode II interlaminar fracture tests. Crack propagations were recorded using a LaVision digital image correlation device, and temperatures were controlled using ThermoStream machine (Thermal Temptronic ATS-545-M-6), which delivers dry air.

Mode I and Mode II interlaminar fracture tests were carried out using double cantilever beam (DCB) and end notch flexure (ENF) samples, shown in Figure 3, prepared according to ASTM D5528 and ASTM D7905/D7905M, respectively. All tests were repeated five times to reduce variation. DCB and ENF tests were conducted in 3 groups, as shown in Table 1. The first group of tests was performed to determine the optimum CNT quantity in CFRP. The DCB and ENF tests were conducted using samples with different CNT densities at constant room temperature and loading rate of 1 mm/s. Once optimum CNT density was determined as 1 g/m^2^, the second group of tests was performed to evaluate the effect of loading rates on the interlaminar fracture toughness. The DCB and ENF tests were conducted using CNT-CFTP with a CNT density of 1 g/m^2^ at a constant room temperature and four different loading rates: 0.001 mm/s, 0.01 mm/s, 0.1 mm/s, and 1 mm/s. The final group of tests was performed to examine the effect of temperature on interlaminar fracture toughness. The DCB and ENF tests were conducted on CNT-CFTP specimens with a CNT density of 1 g/m^2^ at a constant loading rate of 1 mm/s and four different temperatures: −60 °C, −20 °C, 20 °C, and 60 °C.

#### 2.4.1. DCB Experimental Tests

DCB samples with 20 layers were prepared and cut to a specific dimension of 21.68 mm × 132.40 mm × 3.9 mm. Piano hinges were attached to each side of the sample to set the initial crack length to 50 mm from the PTFE film. Before conducting the DCB test, the samples were cured for 24 h. In order to allow visual crack detection during the test, white paint was sprayed on the side of the sample, and a ruler paper was attached to the painted surface. An environmental chamber and ThermoStream machine (Thermal Temptronic ATS-545-M-6) were used for the temperature tests. The piano hinges were loaded into MTS grips, and DCB tests were conducted in tension according to ASTM D5528-13. The DCB test setup is shown in Figure 4.

The compliance calibration method was used for data reduction in order to calculate Mode I interlaminar fracture toughness. At least nine measurements were taken during the test at different crack length values, starting from 60 mm and increasing by 5 mm. The DCB interlaminar fracture toughness is formulated as follows:(1)$$G_{I}=\frac{nP\delta }{2ba},$$
where a is the crack’s length, b is the sample’s width, δ is the displacement, P is the load, and n is the slope of a plot of compliance (δ_i_/P_i_) versus crack length (a_i_) on a log-log scale.

#### 2.4.2. ENF Experimental Tests

ENF samples, which contained 24 layers, were cut into 21.68 mm × 170 mm × 4.1 mm dimensions. During sample preparation, the initial crack length was adjusted to 50 mm for ENF samples. Loading was applied as three-point bending following the ASTM D7905/D7905M standard. The samples were placed on roller supports, ensuring that the distance between the initial crack tip and the left roller was 30 mm. Experimental setup for the ENF test is shown in Figure 5.

Since the compliance calibration method is the only acceptable method for data reduction for the ENF test, it was used as the data reduction method. The ENF sample exhibited unstable crack growth over most of its length; therefore, Mode II interlaminar fracture toughness was obtained as a single value from maximum load points. The strain energy release rate in Mode II interlaminar fracture toughness is formulated as follows:(2)$$G_{II}=\frac{9a^{2}P\delta }{2b\left(2L^{3}+3a^{2}\right)}10^{3},$$
where a is the crack’s length, b is the sample’s width, δ is the displacement, P is the load, and L is the half span length.

### 2.5. Fracture Examination

A Thermofisher Quanta 3D scanning electron microscope was used to examine the CDM samples’ fractured surfaces. The DCB samples were split open after the completion of the tests and imaged 25 mm away from the initial crack tip; thus, we maintained consistency when comparing the samples. The DCB sample’s fractured surface was measured at three different magnifications (100×, 1000×, and 10,000×) to observe micro- and nano-damages in the DCB samples. The SEM location for examining DCB samples is shown in Figure 6.

## 3. Results

Experimental compliance calibrations were performed on DCB and ENF samples at all group tests. Crack propagation in DCB tests–unlike ENF tests–was observed at a stable and almost constant speed during the experiment. The typical fracture mechanism and load–displacement curve of DCB and ENF samples that contain 1 g/m^2^ CNT at 20 °C with a 1 mm/s loading rate is given in Figure 7.

### 3.1. Mode I (DCB) Test

#### 3.1.1. CNT Areal Density Effect

Figure 8 shows the Mode I interlaminar fracture toughness of CNT-CFRP with different CNT areal densities. A non-linear relationship was observed between Mode I interlaminar fracture toughness and CNT areal density. Mode I interlaminar fracture toughness increased by increasing the CNT amount to 1 g/m^2^. Beyond this point, a decrease in Mode I interlaminar fracture toughness was observed. The optimum CNT areal density was determined to be 1 g/m^2^ for the DCB test. This decrease in the fracture toughness is due to the clustering of CNTs in the fiber interface rather than the penetration of individual CNTs into the fibers, leading to a decreased resistance to crack growth [23]. This effect was further confirmed through fracture analysis presented in subsequent sections.

#### 3.1.2. Loading Rate Effect

The experimental results of the DCB samples with 1 g/m^2^ CNT subjected to varying loading rates at room temperature are shown in Figure 9. Mode I interlaminar fracture toughness increased with increasing loading rate. This result is due to the higher loading rates, where CFRP experiences a more rapid crack growth, leading to the formation of more damage-tolerant fracture mechanisms, such as bridging, delamination cracking, or fiber fracture. These mechanisms can increase the energy required to propagate the crack, resulting in increasing interlaminar fracture toughness.

#### 3.1.3. Temperature Effect

Figure 10 shows the interlaminar fracture toughness of DCB samples with 1 g/m^2^ CNT at different temperatures for 1 mm/s loading rate. It is observed that Mode I interlaminar fracture toughness increased with increasing for all temperatures. With the effect of temperature, Mode I interlaminar fracture toughness significantly decreased up to −60 °C. Moreover, with increasing temperature, softness was observed in the epoxy, which might have caused the fibers to be pulled out from the epoxy matrix. This phenomenon is called fiber bridging and increases fracture toughness [36,37]. 

### 3.2. Mode II (ENF) Test

#### 3.2.1. CNT Areal Density Effect

Figure 11 shows CNT-CFRP’s Mode II interlaminar fracture toughness with different CNT areal densities. Enhancement of Mode II fracture toughness in CFRP due to the areal loading of CNTs exhibited a trend similar to that observed in the Mode I experiments. Mode II interlaminar fracture toughness increased with increasing amount of CNTs up to 1 g/m^2^; further increase in amount of CNTs in CFRP resulted in decreasing Mode II fracture toughness. The optimal areal density of CNTs was observed to be 1 g/m^2^ for the ENF test.

#### 3.2.2. Loading Rate Effect

The experimental results of applying varying loading rates on the ENF samples with 1 g/m^2^ CNT at room temperature are shown in Figure 12. It is obvious that Mode II interlaminar fracture toughness increased with increasing loading rate. However, this increase was slightly less in elevated loading rates where the ENF tests were performed.

#### 3.2.3. Temperature Effect

Interlaminar fracture toughness at varying temperatures for the ENF test is shown in Figure 13. The Mode II interlaminar fracture toughness increased with increasing temperature up to 20 °C, then decreased at higher temperatures. Since delamination is smaller in the ENF test, less fiber bridging occurs compared to the corresponding DCB tests. However, the fact that fibers and the matrix have different thermal expansion coefficients introduced microcracks at the matrix–fiber interface, thus resulting in decreased Mode II interlaminar fracture toughness.

To compare the results of all the parameters’ effect on the CNT samples, we present Table 2, showing the change (in percentage) of the resulting values with respect to the reference value for each parameter (temperature, speed, and CNT areal density). It must be noted that, with the minus sign, the value decreased from the reference conditions (20 °C, 1 mm/s, 1 g/m^2^), which is denoted as “REF”. As observed in Table 2, the values of each parameter decreased with decreasing temperature. Those results also proved that 1 g/m^2^ is the ideal quantity of CNTs, since either increasing or decreasing this value corresponded to lower fracture toughness. This follows other studies’ outcomes found in the bibliography [23].

### 3.3. Microstructural Analysis

The fracture surface of DCB test samples was investigated using SEM. The DCB sample’s fractured surface was measured at 100×, 1000×, and 10,000× magnifications to observe micro- and nano-damages in the DCB samples. The fracture surface of the DCB test samples with 1 g/m^2^ CNT-CFRP is shown in Figure 14. The loading rate was 1 mm/s performed at room temperature.

#### 3.3.1. Fracture Analysis of DCB Samples with Varying CNT Areal Density

Figure 15 shows an SEM (100×) of DCB test samples with varying CNT areal density. It can be seen that fiber breakage and fiber imprints decreased with increasing amount of CNT up to 1 g/m^2^; after this limit, a decrease was observed. In addition, fiber breakages were observed where DCB contained no CNT.

Figure 16 shows the fracture surface of the DCB sample with varying CNT areal density at 1000× magnification. Fiber and matrix damage is clearly observed in detail. Specifically, damage in the DCB sample without CNT occurs at the fiber–matrix interface, while the damage in DCB samples to CNT occurs on the matrix surface. The high amounts of CNT in the fiber layers resulted in the fibers banding together at the fiber interface instead of penetrating deep into the fibers. Hence, they demonstrated no resistance to crack growth, resulting in less fracture toughness at a higher than optimum level of CNT areal density.

The fracture surface of the DCB sample with varying CNT areal density at 10,000× magnification is shown in Figure 17.

#### 3.3.2. Fracture Analyses of DCB Samples with Varying Temperature

SEM of DCB test samples that contained 1 g/m^2^ with varying temperatures is shown in Figure 18. It can be seen that the DCB sample exhibited a brittle behavior. With increasing temperature, softness can be seen in the epoxy, which might have caused fiber bridging and fracture toughness to increase.

## 4. Conclusions

The effect of CNT areal density, temperature, and loading rate on Mode I and Mode II interlaminar fracture toughness of carbon nanotubes enhanced with carbon-fiber-reinforced polymers was investigated. The relationship between CNT areal density and Mode I and Mode II fracture toughness was found to be non-linear. Moreover, the optimum CNT amount was 1 g/m^2^, for which the interlaminar strengthening effect is maximum. It has also been observed that toughness in both Modes I and II improved with increasing loading rate. In Mode II deformation, this improvement is slightly less pronounced. Furthermore, Mode I and Mode II interlaminar fracture toughness was observed to be affected differently by varying temperatures. Mode I fracture toughness increased with increasing temperature. Mode II fracture toughness increased by increasing temperature up to room temperature, while at elevated temperatures it decreased. Since CNTs, resins, and fibers shrink at different rates, stress will be generated within the fiber, matrix, and fiber–matrix interface. This stress has the potential to inflict damage upon the composite laminates. Additionally, prolonged exposure to low temperatures can cause epoxy resins to become brittle, deteriorating interlaminar fracture toughness.

## Figures and Tables

**Figure 1 nanomaterials-13-01729-f001:**
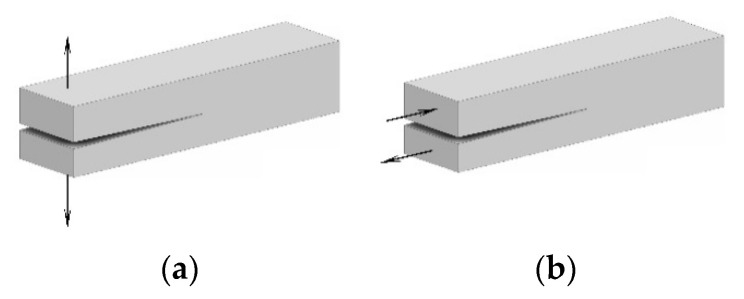
Interlaminar Fracture Loading modes: (**a**) DCB; (**b**) ENF.

**Figure 2 nanomaterials-13-01729-f002:**
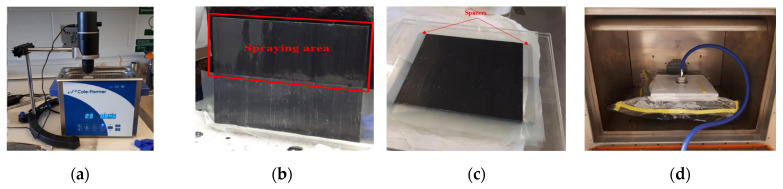
Sample preparation of (**a**) dispersing CNT–ethanol solutions, (**b**) spraying CNT–ethanol solutions, (**c**) stacking up laminates and glass, and (**d**) curing laminates.

**Figure 3 nanomaterials-13-01729-f003:**
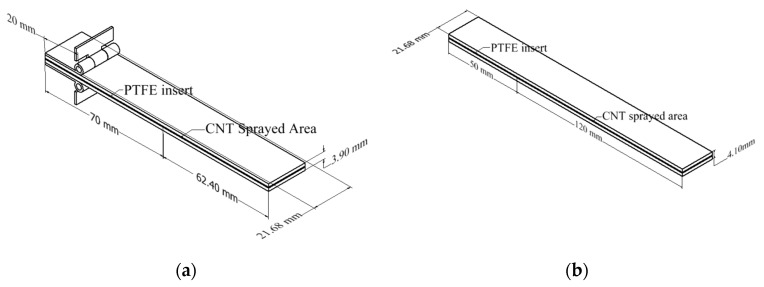
Sample dimensions for interlaminar fracture tests of: (**a**) Mode I and (**b**) Mode II.

**Figure 4 nanomaterials-13-01729-f004:**
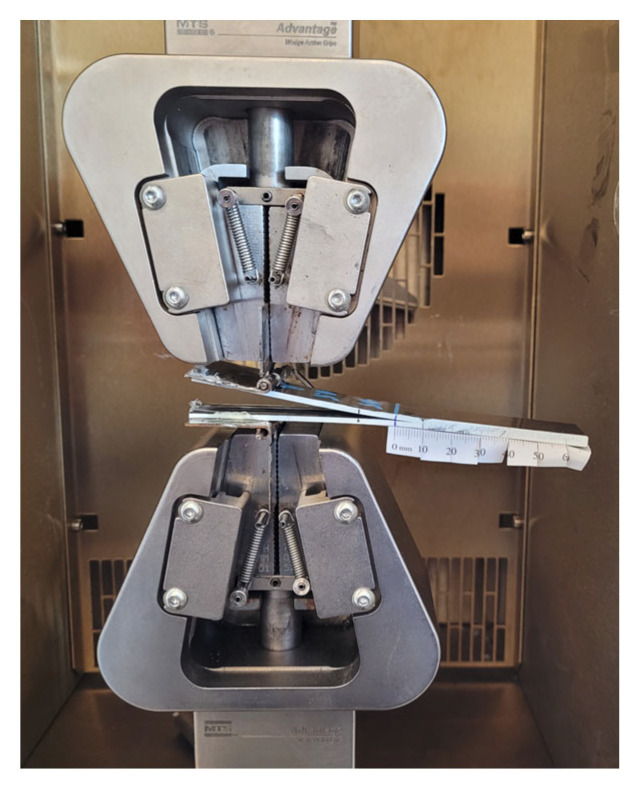
DCB test setup.

**Figure 5 nanomaterials-13-01729-f005:**
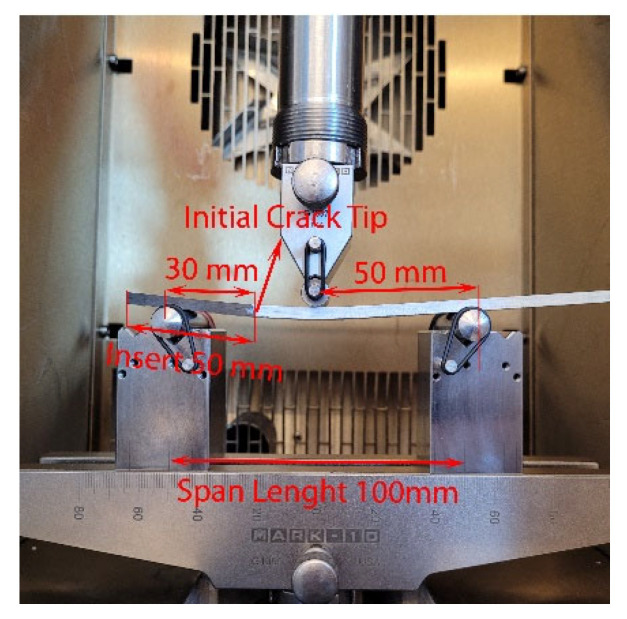
ENF test setup.

**Figure 6 nanomaterials-13-01729-f006:**
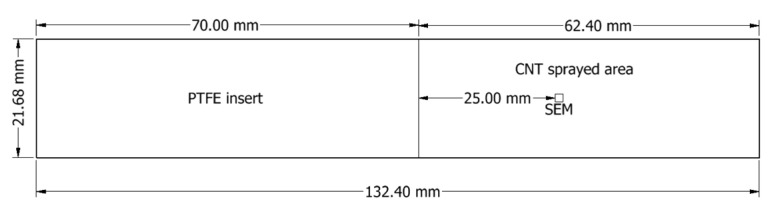
SEM location for testing DCB samples.

**Figure 7 nanomaterials-13-01729-f007:**
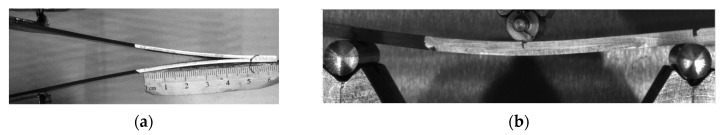
Fracture mechanism of (**a**) DCB test and (**b**) ENF test; the corresponding load–displacement curves of (**c**) DCB test and (**d**) ENF test.

**Figure 8 nanomaterials-13-01729-f008:**
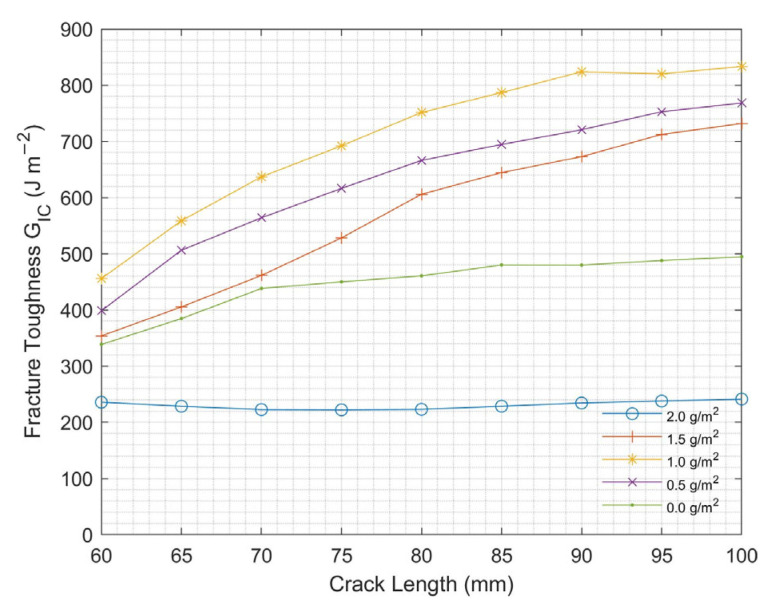
Mode I fracture toughness at varying CNT areal density for DCB test.

**Figure 9 nanomaterials-13-01729-f009:**
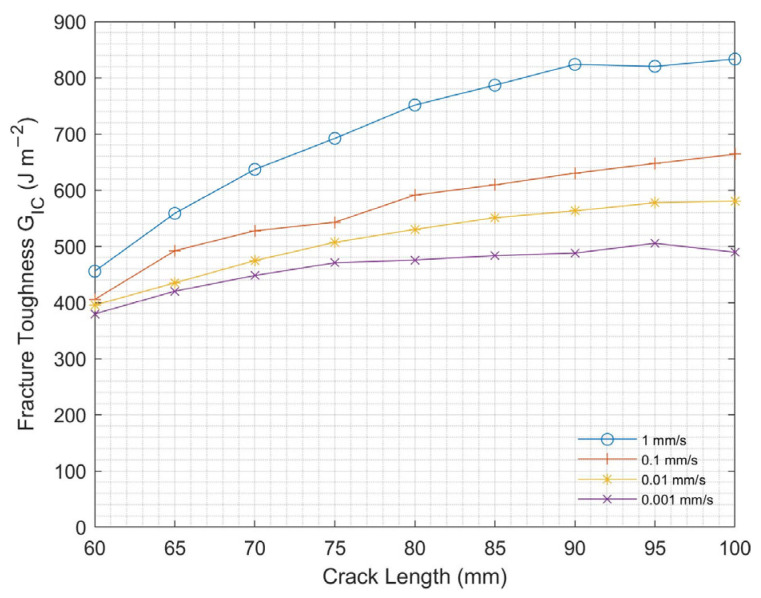
Mode I fracture toughness at varying loading rates for DCB test.

**Figure 10 nanomaterials-13-01729-f010:**
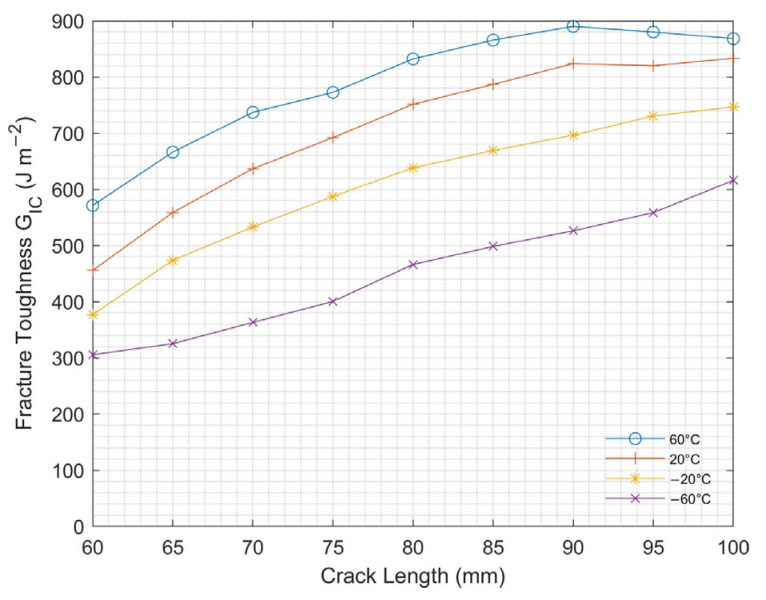
Mode I fracture toughness at varying temperatures for DCB test.

**Figure 11 nanomaterials-13-01729-f011:**
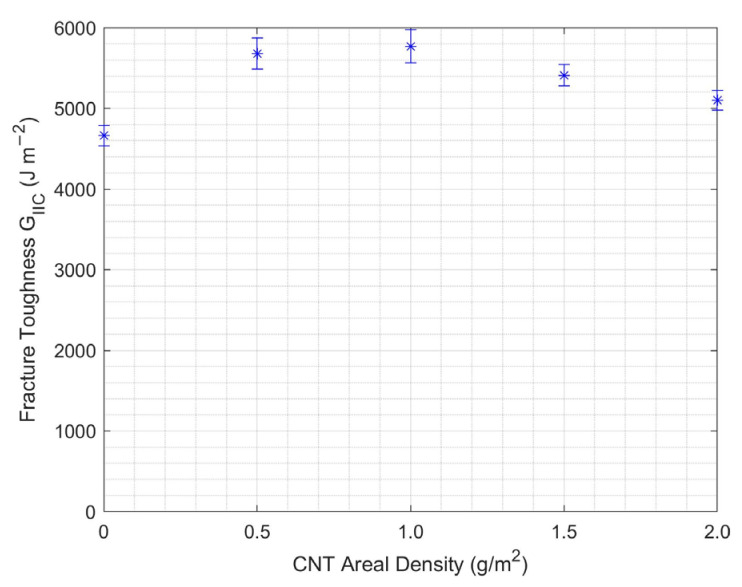
Mode II fracture toughness at varying CNT areal density for the ENF test.

**Figure 12 nanomaterials-13-01729-f012:**
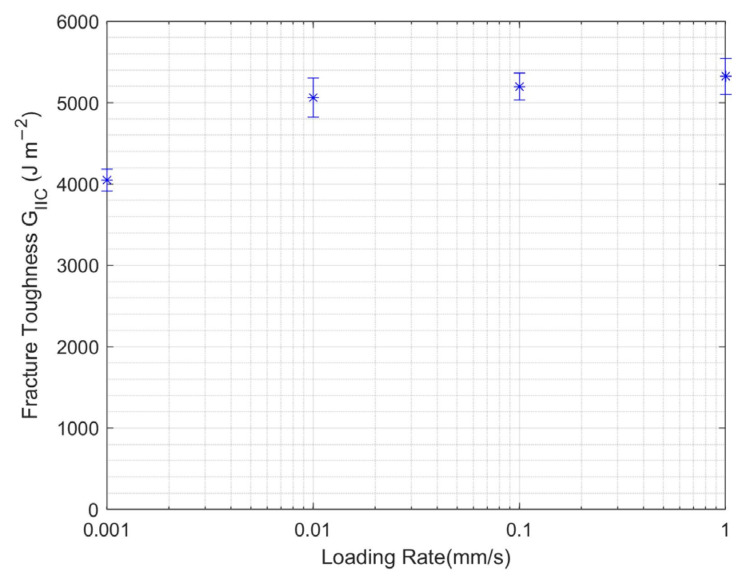
Mode II fracture toughness at varying loading rates for ENF test.

**Figure 13 nanomaterials-13-01729-f013:**
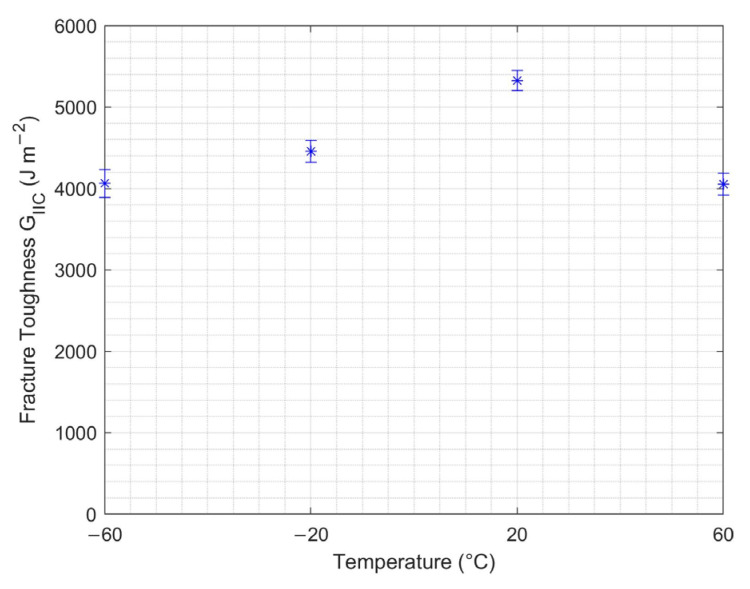
Mode II fracture toughness at varying temperatures for ENF test.

**Figure 14 nanomaterials-13-01729-f014:**
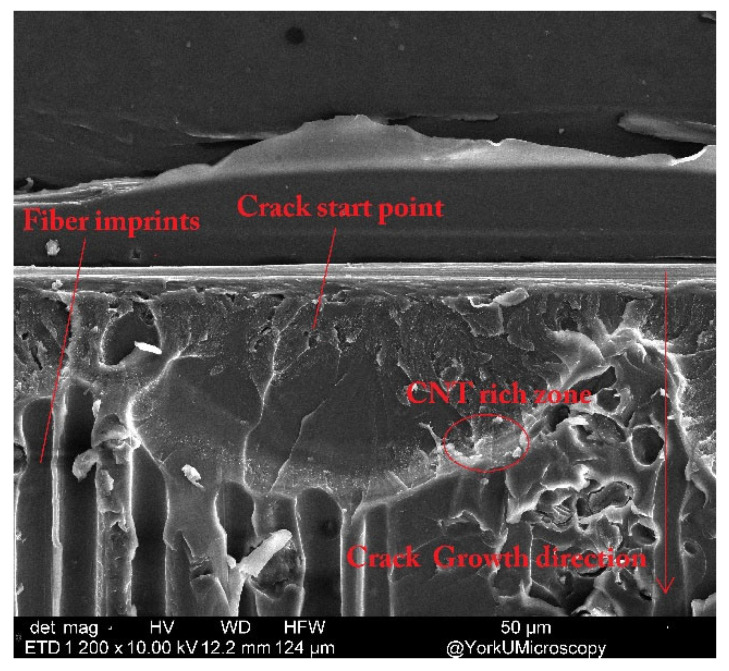
SEM (1000×) image of the fracture surface of the DCB sample.

**Figure 15 nanomaterials-13-01729-f015:**
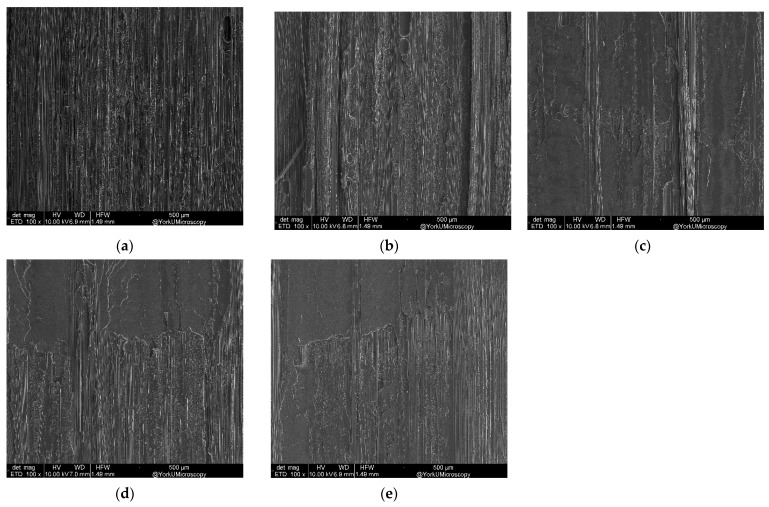
SEM (100×) for varying CNT areal density g/m^2^: (**a**) 0, (**b**) 0.5, (**c**) 1.0, (**d**) 1.5 and (**e**) 2.

**Figure 16 nanomaterials-13-01729-f016:**
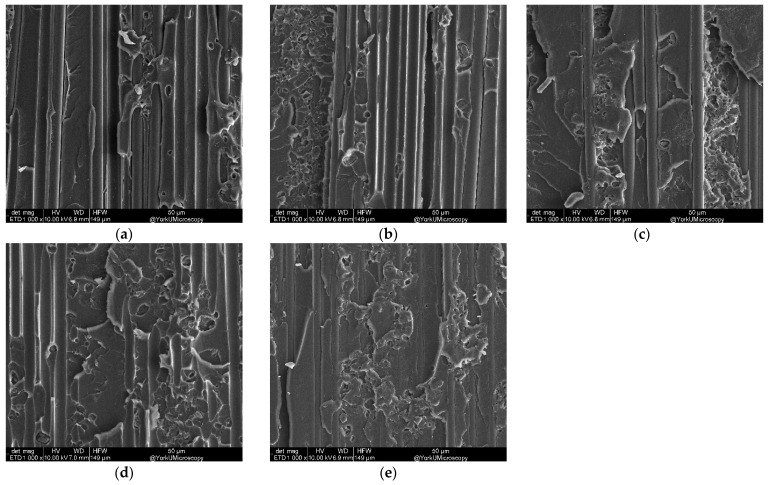
SEM (1000×) for varying CNT areal density g/m^2^: (**a**) 0, (**b**) 0.5, (**c**) 1.0, (**d**) 1.5, and (**e**) 2.

**Figure 17 nanomaterials-13-01729-f017:**

SEM (10,000×) for varying CNT areal density g/m^2^: (**a**) 0, (**b**) 0.5, (**c**) 1.0, (**d**) 1.5, and (**e**) 2.

**Figure 18 nanomaterials-13-01729-f018:**
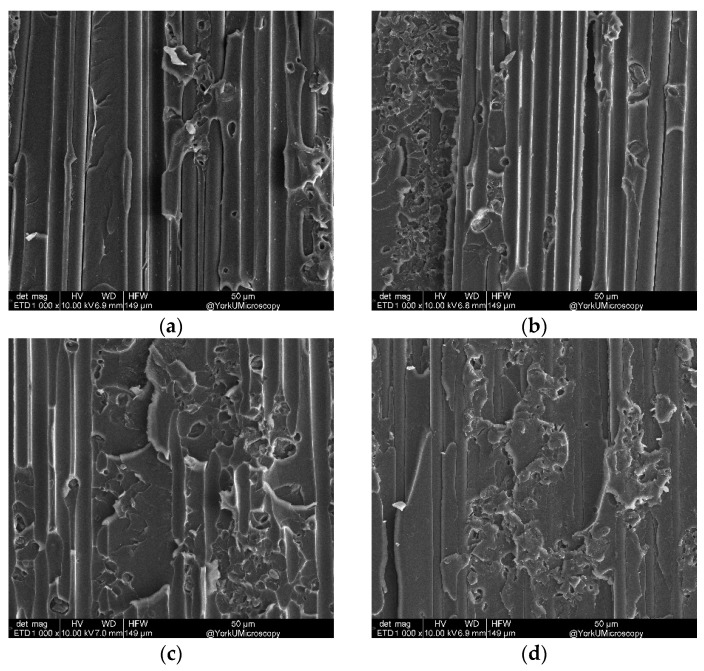
SEM for 1000× magnification and temperature: (**a**) −60 °C, (**b**) −20 °C, (**c**) 20 °C, and (**d**) 60 °C.

**Table 1 nanomaterials-13-01729-t001:** Type and parameters of tests conducted.

Test Groups (Objective)	Constant Values	Variables
1—CNT Density (g/m^2^)	20 °C, 1 mm/s	0, 0.5, 1, 1.5, 2 [g/m^2^]
2—Loading Rate (mm/s)	20 °C, 1 g/m^2^	0.001, 0.01, 0.1, 1 [mm/s]
3—Temperature	1 mm/s, 1 g/m^2^	−60, −20, 20, 60 [°C]

**Table 2 nanomaterials-13-01729-t002:** Changes of fracture toughness (%) from the reference values of each parameter.

Temperature (°C)	Change (%)	Speed (mm/s)	Change (%)	Areal Density (g/m^2^)	Change (%)
−60	−26%	0.001	−39%	0	−40.7%
−20	−10%	0.01	−30%	0.5	−7.8%
20	REF	0.1	−20%	1	REF
60	7%	1	REF	1.5	−12.2%
				2	−62.1%

## Data Availability

Not applicable.

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
