# Peer review of "Characterization of Mode I and Mode II Interlaminar Fracture Toughness in CNT-Enhanced CFRP under Various Temperature and Loading Rates"

_nanomaterials, 2023, doi:10.3390/nano13111729_

Round 1
Reviewer 1 Report
1. The novelty of this work should be identified more clearly in the introduction part.
2. Does the direction of carbon fiber affects the results? How?
Reviewer 2 Report
Comments to nanomaterials-2384283-peer-review-v1:
This study investigates the influence of temperature and loading rate on the Mode I and Mode II interlaminar fracture behavior of carbon nanotubes enhanced carbon fiber-reinforced pol-ymer (CNT-CFRP). CNT-induced toughening of the epoxy matrix is characterized by producing CFRP with varying loading of CNT areal density. Moreover, it was found that CNT-CFRP fracture toughness increased linearly with the loading rate in Mode I and Mode II. On the other hand, different responses to changing temperature were observed; Mode I fracture toughness increased when elevating the temperature, while Mode II fracture toughness increased with increasing temperature up to room temperature and decreased at higher temperatures. There are still some details in the manuscript that need to be further supplemented, the comments are given below:
1. The influence mechanism of CNT dosage, loading rate and temperature on fracture toughness of the composites should be further supplemented.
2. How to distinguish between carbon fibers and CNT in Figures 15 and 16.
3. Please further highlight the novelty of the article. In this work, the authors reported CNT-CFRP composites. The authors are encouraged to tell the main advantages of Carbon fibers from other fibers, and the following works about quartz fibers (Journal of Materials Science & Technology, 2023, 139: 189), glass fibers (Journal of Materials Science & Technology, 2021, 75: 225) and PBO fibers (Chinese Journal of Chemistry, 2023, 41: 939; Journal of Materials Science & Technology, 2023, 152: 16; Science Bulletin, 2022, 67(21): 2196), etc, may be for your description, to reveal the advantages or novelty of your work, finally to further highlight the theme.
4. How to demonstrate that the epoxy resin in Figure 18 becomes soft.
5. There are many errors in the manuscript, for example:there are two consecutive periods on line 103; the four pictures of Figure 2 should be the same size.
There are some grammar and word mistakes in the manuscript. Please go through the manuscript carefully again.
Reviewer 3 Report
The paper is noteworthy and presents very current engineering issues using current research techniques. The paper is an experimental study, where research based on the application of interdisciplinary research techniques is clearly exposed. In order for the paper to be published, it is necessary to make certain changes:
1. In the introduction, please clearly demonstrate the novelty of the present work in relation to other thematically similar research works.
2. in the introduction or a stage of the research description, please refer to works related to the study of thin-walled composite structures (DOI: 10.12913/22998624/67677 and 10.1016/j.compstruct.2020.112388) and to paper in which similar research (DOI: 10.1088/1757-899X/416/1/012055)was presented.
3. The conclusions are described in terms of qualitative evaluation and not quantitative. Please describe them from a quantitative point as well.
Reviewer 4 Report
The article is interesting and deals with a topic of interest for the journal. In this reviewer's opinion, revisions are necessary, as only a few points need to be clarified or improved to make the work better.
In the introduction, the effect of nanotube dispersion and how they can be dispersed in polymer matrices are marginally addressed. It would also be useful to discuss the use not only of mixed composites but also of hierarchical reinforcement in polymer composites where nanotubes are grafted to carbon fibers, resulting in improved fracture toughness.
In the case of the temperature function test, it is unclear whether there is a correlation between the polymer's Tg and its effect on the reinforcement used, in addition to the expansion of fibers (which is actually minimal).
Were no pull-out effects observed in the other two cases? It seems strange from the images that the fibers did not break, even though the toughness effect provided so many advantages.
It would be useful to observe the imperfect dispersion of nanotubes inside the polymer using SEM to confirm that it is indeed a genuine defect point of the composite as shown in many article in literature[1–3].
There are numerous typing errors throughout the text, such as line 85, line 103, and line 147. I recommend a careful proofreading to eliminate these issues as much as possible.
Reference
1. Lavagna, L.; Marchisio, S.; Merlo, A.; Nisticò, R.; Pavese, M. Polyvinyl Butyral‐based Composites with Carbon Nanotubes: Efficient Dispersion as a Key to High Mechanical Properties. Polymer Composites 2020, pc.25661, doi:10.1002/pc.25661.
2. Thostenson, E.T.; Ren, Z.; Chou, T.-W. Advances in the Science and Technology of Carbon Nanotubes and Their Composites: A Review. Composites Science and Technology 2001, 61, 1899–1912, doi:10.1016/S0266-3538(01)00094-X.
3. Coleman, J.N.; Khan, U.; Gun’ko, Y.K. Mechanical Reinforcement of Polymers Using Carbon Nanotubes. Advanced Materials 2006, 18, 689–706, doi:10.1002/adma.200501851.
Round 2
Reviewer 1 Report
Accept
Author Response
Thank you very much
Reviewer 2 Report
Comments to nanomaterials-2384283-peer-review-v1:
This study investigates the influence of temperature and loading rate on the Mode I and Mode II interlaminar fracture behavior of carbon nanotubes enhanced carbon fiber-reinforced pol-ymer (CNT-CFRP). CNT-induced toughening of the epoxy matrix is characterized by producing CFRP with varying loading of CNT areal density. Moreover, it was found that CNT-CFRP fracture toughness increased linearly with the loading rate in Mode I and Mode II. On the other hand, different responses to changing temperature were observed; Mode I fracture toughness increased when elevating the temperature, while Mode II fracture toughness increased with increasing temperature up to room temperature and decreased at higher temperatures. There are still some details in the manuscript that need to be further supplemented, the comments are given below:
1. The influence mechanism of CNT dosage, loading rate and temperature on fracture toughness of the composites should be further supplemented.
2. How to distinguish between carbon fibers and CNT in Figures 15 and 16.
3. Please further highlight the novelty of the article. In this work, the authors reported CNT-CFRP composites. The authors are encouraged to tell the main advantages of Carbon fibers from other fibers, and the following works about quartz fibers (Journal of Materials Science & Technology, 2023, 139: 189), glass fibers (Journal of Materials Science & Technology, 2021, 75: 225) and PBO fibers (Chinese Journal of Chemistry, 2023, 41: 939; Journal of Materials Science & Technology, 2023, 152: 16; Science Bulletin, 2022, 67(21): 2196), etc, may be for your description, to reveal the advantages or novelty of your work, finally to further highlight the theme.
4. How to demonstrate that the epoxy resin in Figure 18 becomes soft.
5. There are many errors in the manuscript, for example:there are two consecutive periods on line 103; the four pictures of Figure 2 should be the same size.
Author Response
Thank you very much for your comments.
The comments and suggestions listed below were addressed in the first round.
In this round, we made minor changes to the manuscript.
Reviewer 4 Report
The authors responded comprehensively to all the issues raised. The article is recommended for publication.
Author Response
Thank you very much for your comments.